# Inhibition of Tyrosyl-DNA Phosphodiesterase 1 by Lipophilic Pyrimidine Nucleosides

**DOI:** 10.3390/molecules25163694

**Published:** 2020-08-13

**Authors:** Alexandra L. Zakharenko, Mikhail S. Drenichev, Nadezhda S. Dyrkheeva, Georgy A. Ivanov, Vladimir E. Oslovsky, Ekaterina S. Ilina, Irina A. Chernyshova, Olga I. Lavrik, Sergey N. Mikhailov

**Affiliations:** 1Institute of Chemical Biology and Fundamental Medicine, Siberian Branch of the Russian Academy of Sciences, 8 Lavrentiev Ave., 630090 Novosibirsk, Russia; a.zakharenko73@gmail.com (A.L.Z.); elpida80@mail.ru (N.S.D.); katya.plekhanova@gmail.com (E.S.I.); chernyshova0305@gmail.com (I.A.C.); 2Engelhardt Institute of Molecular Biology, Russian Academy of Sciences, 32 Vavilova Str., 119991 Moscow, Russia; mdrenichev@mail.ru (M.S.D.); georgyivanovk423@gmail.com (G.A.I.); vladimiroslovsky@gmail.com (V.E.O.); 3Department of Natural Sciences, Novosibirsk State University, 2 Pirogova Str., 630090 Novosibirsk, Russia

**Keywords:** nucleosides, DNA repair, tyrosyl-DNA phosphodiesterase, Tdp1 inhibition, topotecan

## Abstract

Inhibition of DNA repair enzymes tyrosyl-DNA phosphodiesterase 1 and poly(ADP-ribose)polymerases 1 and 2 in the presence of pyrimidine nucleoside derivatives was studied here. New effective Tdp1 inhibitors were found in a series of nucleoside derivatives possessing 2′,3′,5′-tri-*O*-benzoyl-d-ribofuranose and 5-substituted uracil moieties and have half-maximal inhibitory concentrations (IC_50_) in the lower micromolar and submicromolar range. 2′,3′,5′-Tri-*O*-benzoyl-5-iodouridine manifested the strongest inhibitory effect on Tdp1 (IC_50_ = 0.6 μM). A decrease in the number of benzoic acid residues led to a marked decline in the inhibitory activity, and pyrimidine nucleosides lacking lipophilic groups (uridine, 5-fluorouridine, 5-chlorouridine, 5-bromouridine, 5-iodouridine, and ribothymidine) did not cause noticeable inhibition of Tdp1 (IC_50_ > 50 μM). No PARP1/2 inhibitors were found among the studied compounds (residual activity in the presence of 1 mM substances was 50–100%). Several *O*-benzoylated uridine and cytidine derivatives strengthened the action of topotecan on HeLa cervical cancer cells.

## 1. Introduction

DNA repair systems are resistant to the action of chemotherapeutic drugs and ionizing radiation, and therefore the beneficial effect of antitumor therapy depends on the effectiveness of DNA repair systems in tumor cells. Accordingly, targeted inhibition of DNA repair enzymes can increase the effectiveness of chemotherapeutic drugs used in clinical practice and may enable reducing their dose, which in turn can decrease the overall toxicity of treatment [1,2,3,4,5,6,7,8]. A number of enzymes and protein factors are involved in DNA repair [9,10,11,12,13,14,15]. Poly(ADP-ribose)polymerase 1 (PARP1, EC 2.4.2.30)—a key regulator of DNA repair mechanisms—is the most studied pharmacological target for the creation of targeted drugs [16]. Four PARP1 inhibitors (olaparib, rucaparib, niraparib, and talazoparib) are used in clinical practice for the treatment of ovarian cancer.

Among promising targets is the enzyme tyrosyl-DNA phosphodiesterase 1 (Tdp1), which is considered an important target for the antitumor therapy based on topoisomerase 1 (Top1) inhibitors [17]. Tdp1 plays a key role in the removal of Top1–DNA adducts stabilized by Top1 inhibitors such as camptothecin and its clinically relevant derivatives. The catalytic action of Tdp1 is the hydrolysis of the phosphotyrosyl bond between the 3′ end of DNA and a tyrosine residue of Top1 in a stabilized Top1–DNA complex, with hydrophobic interactions playing an important part in the binding of Tdp1 to the Top1–DNA complex [18]. The literature describes Tdp1 inhibitors of various structures based on different classes of natural and synthetic compounds [3,4]; these inhibitors have half-maximal inhibitory concentrations (IC_50_) in the range 0.015–10.000 µM. Recently, a novel group of Tdp1 inhibitors effective in the submicromolar range of concentrations was found among disaccharide nucleosides with lipophilic groups [19]. The current work represents a continuation of our earlier studies on the inhibition of DNA repair enzymes by nucleoside compounds [19,20,21,22,23,24].

## 2. Results

### 2.1. DNA Repair Enzyme Inhibition

To study the inhibition of the DNA repair enzyme by nucleoside derivatives, a number of 5-substituted derivatives of 2′,3′,5′-tri-*O*-benzoyluridine were synthesized according to the procedures elaborated earlier by Vorbrüggen and coworkers [25] and by Prasad and coworkers [26] (Appendix A).

Previously, we designed a real-time oligonucleotide biosensor based on the capability of Tdp1 to remove fluorophore quenchers from the 3′ end of DNA [27]. The single-stranded substrate was a 16-mer oligonucleotide containing both a 5′-FAM fluorophore donor and a quenching 3′-BHQ moiety. When the quencher is removed by Tdp1, FAM fluorescence flares up and can be detected by a fluorimeter. Fluorescence intensity diminishes in the presence of an inhibitor; curves of the obtained Tdp1 activity versus inhibitor concentration can be used to calculate IC_50_ values. The results of the Tdp1 assay for nucleoside derivatives **1** and **2** are shown in Table 1. We also investigated some structurally related compounds obtained by previously developed methods [25,26,28]; these compounds feature a modification either in a carbohydrate or heterocyclic moiety (Table 1).

In addition to the effects on Tdp1, we tested the effects of the lipophilic nucleosides on DNA break sensors poly(ADP-ribose)polymerases 1 and 2 (PARP1 and PARP2), in a test system based on tritium-labeled NAD^+^ [20]. None of the tested nucleosides caused significant inhibition of either PARP1 or PARP2 (residual activity in the presence of 1 mM substances was 50–100%, data not shown).

### 2.2. Cytotoxicity and Sensitization of Tumor Cells to the Effect of Topotecan

#### 2.2.1. Cytotoxicity

We evaluated the cytotoxicity of compounds that showed a Tdp1-inhibitory activity as well as their impact on the cytotoxic effect of topotecan. First, we assessed the toxicity of the compounds to HeLa cells (cervical cancer cells). Analysis of the intrinsic cytotoxicity of the compounds was examined in the EZ4U Cell Proliferation and Cytotoxicity Assay (Biomedica, Austria). The tested compounds were added to the medium (the volume of added reagents was 1/100 of the total volume of the culture medium, and the amount of DMSO (dimethyl sulfoxide) was 1% of the final volume), and the cell culture was monitored for 3 days. Control cells were grown in the presence of 1% DMSO. Intrinsic cytotoxicity of the compounds was either absent or negligible at concentrations up to 100 μM (>80% live cells) with the exception of compounds **2b**, **2c**, and **7**, in the presence of which, at a concentration of 100 μM, 30–45% of the cells died (Table 1 and Figure 1).

#### 2.2.2. Tumor Cell Sensitization to Topotecan

To study the influence of the inhibitors on the cytotoxic effect of topotecan, the latter from the manufacturer ACTAVIS GROUP PTC ehf. was used. To select a nontoxic but effective concentration of the nucleoside compounds, we investigated their cytotoxicity in the presence of 2 μM topotecan (concentration causing the death of half the HeLa cells). Three independent cytotoxicity assays were performed with each inhibitor in combination with topotecan. For the cells treated with a nucleoside derivative alone, values in the control wells treated with 1% DMSO were set to 100% viability. For the cells treated with a combination of the drugs, the viability of cells treated only with topotecan was set to 100%. Topotecan significantly (*p* = 0.05, Mann–Whitney *U* test) enhanced the cytotoxicity of compounds **2g**, **7**, and **8** (Figure 2a). For compound **5**, the increase in cytotoxicity in the presence of topotecan was significant only at the maximum concentration, 100 μM. For compound **6**, the increase in cytotoxicity in the presence of topotecan (Figure 2a, cyan bars) was statistically insignificant. The cytotoxicity of the remaining compounds in the presence of topotecan did not change significantly (Figure 2b).

For the three compounds **2g**, **7**, and **8** that had a synergistic effect with topotecan, a nontoxic concentration, 20 μM, which exerted a noticeable synergistic effect, was chosen for further experiments on HeLa cells at various doses of topotecan (Figure 3).

For **5**, a low cytotoxic concentration, 50 μM, was chosen because at a concentration of 20 μM, no synergy with topotecan was observed (Figure 2a, yellow bars). Compound **5** had a weak sensitizing effect on topotecan-treated cells at high concentrations (a significant effect at 10 μM; Figure 3, the orange graph compared to the black one). Compounds **2g**, **7**, and **8** doubled the cytotoxicity of topotecan.

## 3. Discussion

It has been shown earlier that disaccharide nucleosides with lipophilic groups inhibit Tdp1 in a submicromolar range of concentrations and are only weakly toxic to cancerous and healthy cells [19]. On the other hand, disaccharide nucleosides tend to inhibit PARP1 [20,22], and this activity may reduce their selectivity toward Tdp1 in the cell. Therefore, we focused on nucleoside inhibitors of Tdp1, which are structurally modified previously described disaccharide inhibitors of Tdp1. New efficient Tdp1 inhibitors were found in a series of ribofuranose nucleoside derivatives and have IC_50_ values in the low micromolar and submicromolar range (Table 1). It was demonstrated here that for the manifestation of an inhibitory effect, the presence of the 2,3,5-tri-*O*-benzoyl-β-d-ribofuranose (2,3,5-tri-*O*-Bz-β-d-Rib) residue is necessary. Pyrimidine nucleosides (uridine, 5-fluorouridine, 5-chlorouridine, 5-bromouridine, 5-iodouridine, and ribothymidine) without lipophilic groups did not inhibit Tdp1 (IC_50_ > 50 μM).

According to the calculations of the coefficient of distribution (logP) between the 1-octanol-water phases, the introduction of benzoyl groups (Bz) into pyrimidine nucleosides led to an increase in logP and therefore could improve their penetration of cellular and nuclear membranes (Table 1). The magnitude of Tdp1 inhibition by the modified uridine nucleosides linearly correlated with the increase in logP (Figure 4 and Appendix A).

The presence of the bulky 2,3,5-tri-*O*-Bz-β-d-Rib residue significantly enhanced the inhibitory activity (IC_50_ = 6.3 μM for 2′,3′,5′-tri-*O*-benzoyluridine **2a** and 0.6 μM for 2′,3′,5′-tri-*O*-benzoyl-5-iodouridine **2d**) in comparison with nucleoside derivatives lacking benzoyl groups (Table 1). 1-*O*-acetyl-2,3,5-tri-*O*-Bz-β-d-Rib (**9**) exerted a Tdp1-inhibitory activity similar to that of *O*-benzoylated cytidine (**8**) derivatives and a higher Tdp1-inhibitory activity than that of *O*-benzoylated uridine (**2a**), 5-fluorouridine (**2b**), and pyrimidone (**5** and **6**) derivatives (Table 1). Most of these compounds were found to be characterized by lower logP than that of ribofuranose derivative **9** (Table 1, logP = 5.62). *O*-benzoylated uridine derivatives with logP comparable to or higher than 5.62 manifested a higher Tdp1-inhibitory activity in comparison with a ribofuranose derivative (**9**). A decrease in the number of benzoic acid residues caused a marked decline in the inhibitory activity (Figure 5, IC_50_ = 23 μM for 2′,3′-di-*O*-benzoyluridine **4a** and IC_50_ > 100 μM for 5′-*O*-benzoyluridine **3a**).

Thus, it can be claimed that lipophilicity and the number of benzoyl groups in the modified nucleoside compounds strongly influence Tdp1 inhibition.

It has been previously reported that disaccharide nucleosides [20,22] and nucleoside derivatives with modified nitrogen bases [29,30] inhibit PARP1. Therefore, in addition to the effect on Tdp1, a possible effect of the lipophilic nucleosides on DNA break sensors PARP1 and PARP2 was tested. Our compounds had virtually no impact on the activity of either enzyme (residual activity in the presence of each 1 mM substance was 50–100%), and therefore may be employed for the development of selective nucleoside inhibitors of Tdp1.

Several papers have been published in the past decade on the applicability of Tdp1 inhibitors for the sensitization of tumors to the action of anticancer topoisomerase 1 inhibitors (e.g., topotecan) see reviews [3,4,17]. Therefore, on HeLa cells, we also studied the effect of the synthesized Tdp1 inhibitors in combination with topotecan, which is the Top1 poison used in clinical practice. Given that Tdp1 inhibitors are supposed to be used in therapeutic cocktails, it is important that they do not exert their own toxicity and do not enhance the existing adverse effects of cancer therapy. According to the experiments on the viability of HeLa cells in the presence of the synthesized pyrimidine lipophilic nucleosides, the intrinsic cytotoxicity of the compounds was either absent or negligible at concentrations up to 100 μM (Figure 1). The enhanced cytotoxicity of topotecan in the presence of the synthesized compounds was then studied. Four nucleoside compounds (**2g**, **5**, **7**, and **8**) each enhanced the cytotoxicity of a 2 μM topotecan solution (Figure 2a), while the cytotoxicity of topotecan in the presence of the other compounds did not significantly change (Figure 2b). It was shown that compounds **7** (containing a 6-methoxy-2-oxopyrimidine residue), **2g** (containing a 6-methyluracil residue), and cytosine derivative **8** at a nontoxic concentration, 20 μM, have a sensitizing effect on topotecan-treated cells, thereby doubling topotecan’s activity toward HeLa cells (Figure 3). These results suggest that nucleoside compounds containing a 2,3,5-tri-*O*-Bz-β-d-Rib group and a uracil base modified at positions 4 or 6 can penetrate into a cell nucleus and accumulate in it.

Tdp1 was discovered because of its ability to hydrolyze the stalled covalent 3′-phospho-tyrosyl that represents the chemical linkage between the active-site tyrosine of DNA topoisomerase I (Topo1) and DNA [31,32,33]. The list of substrates of Tdp1 has since grown and consists of protein–DNA adducts, such as camptothecin-stabilized Topo1–DNA adducts, and modified nucleotides, including oxidized nucleotides, apurinic/apyrimidinic sites, and chain-terminating nucleoside analogs reviewed in [17] and described in [18,34]. Because many substrates of Tdp1 arise when anticancer and antiretroviral drugs are used, this enzyme is becoming an important therapeutic target. In addition, a natural catalytic-site Tdp1 mutant increases the stability of a Tdp1–DNA reaction intermediate and forms the molecular basis for an autosomal recessive neurodegenerative disease called spinocerebellar ataxia with axonal neuropathy (SCAN1).

A number of studies confirmed the hypothesis that Tdp1 is responsible for the resistance of some types of cancer to these drugs [35]: Tdp1-deficient human cells are hypersensitive to camptothecin [36,37,38]. Furthermore, recent data indicate that suppression of Tdp1 expression by minocycline enhances the antimetastatic effect of irinotecan and increases the life span of experimental animals [39]. Conversely, in cells with increased Tdp1 expression, camptothecin and etoposide cause less DNA damage [40,41]. Moreover, the response to irinotecan therapy is lower in tumors of the intestine with overexpression of Tdp1 [42]. Thus, it is expected that a combination of anticancer drugs and Tdp1 inhibitors can significantly increase the effectiveness of chemotherapy.

## 4. Materials and Methods

### 4.1. General

The solvents and materials were reagent grade and were used without additional purification. Column chromatography was performed on silica gel (Kieselgel 60, 0.040–0.063 mm, Merck, Darmstadt, Germany. Thin-layer chromatography was carried out on Alugram SIL G/UV254 (Macherey-Nagel, Düren, Germany) with UV visualization. Melting points were determined by means of Electrothermal Melting Point Apparatus IA6301 and are uncorrected. ^1^H and ^13^C (with complete proton decoupling) NMR spectra were recorded on a Bruker AMX 400 NMR instrument (Bruker, Billerica, MA, USA). ^1^H-NMR-spectra were acquired at 400 MHz, and ^13^C-NMR spectra at 100 MHz. Chemical shifts in parts per million (ppm) were measured relative to the residual solvent signals as internal standards (CDCl_3_, ^1^H: 7.26 ppm, ^13^C: 77.1 ppm; DMSO-*d*_6_, ^1^H: 2.50 ppm, and ^13^C: 39.5 ppm). Spin–spin coupling constants (*J*) are given in Hertz (Hz). The following compounds were prepared according to the methods reported earlier: compounds **2a**–**e** [26], **2f** [43], compound **2g** [44], and derivatives of pyrimidine-4-one, pyrimidine-2-one, 2-oxo-4-methoxypyrimidine (**5**–**7**) [25,44,45], and of 2′,3′,5′-tri-*O*-benzoylcytidine (**8**) [26]. The compounds’ coefficients of partition between the octanol–water phases (logP) were calculated in the Instant J. Chem. (ChemAxon^®^) software (Budapest, Hungary).

### 4.2. Real-Time Detection of Tdp1 Activity

This procedure was reported in our previous work [27]. The approach consists of fluorescence intensity measurement in a reaction of quencher removal from a fluorophore quencher–coupled DNA oligonucleotide catalyzed by Tdp1 in the presence of an inhibitor (the control samples contained 1% of DMSO). The reaction mixtures (200 μL) contained Tdp1 buffer (50 mM Tris-HCl pH 8.0, 50 mM NaCl, and 7 mM β-mercaptoethanol), 50 nM biosensor, an inhibitor being tested, and purified Tdp1 (1.5 nM). The biosensor was a single-stranded oligonucleotide (5′-[FAM] AAC GTC AGGGTC TTC C [BHQ]-3′) containing a fluorophore at the 5′ end (5,6-FAM) and Black Hole Quencher 1 (BHQ) at the 3′ end and was synthesized in the Laboratory of Biomedical Chemistry at the Institute of Chemical Biology and Fundamental Medicine (Novosibirsk, Russia).

The reactions were incubated at a constant temperature of 26 °C on a POLARstar OPTIMA fluorimeter (BMG LABTECH, GmbH, Ortenberg, Germany) to measure fluorescence every 55 s (ex. 485/em. 520 nm) during the linear phase (here, data from minute 0 to minute 8). The values of IC_50_ were determined using a six-point concentration response curve in three independent experiments and were calculated using MARS Data Analysis 2.0 (BMG LABTECH).

### 4.3. Cell Culture Assays

Intrinsic toxicity of the compounds to human cell line HeLa (cervical cancer) was examined by means of the EZ4U Cell Proliferation and Cytotoxicity Assay (Biomedica, Wien, Austria), according to the manufacturer’s protocols. The cells were grown in Iscove’s modified Dulbecco’s medium (IMDM) with 40 μg/mL gentamicin, 50 IU/mL penicillin, 50 μg/mL streptomycin (MP Biomedicals), and 10% of fetal bovine serum (Biolot) in a 5% CO_2_ atmosphere. After formation of a 30–50%-confluent monolayer, one of the tested compounds was added to the medium (the volume of the added reagents was 1/100 of the total volume of the culture medium, and the amount of DMSO was 1% of the final volume), and the cell culture was monitored for 3 days. Control cells were grown in the presence of 1% DMSO. To assess the influence of the inhibitors on the cytotoxic effect of topotecan, the latter from the manufacturer ACTAVIS GROUP PTC ehf. (Bucharest, Romania) was used. First of all, 50% cytotoxic concentrations of topotecan and of each inhibitor were determined to attain a defined single-agent effect. Second, three independent tests were performed with each inhibitor in combination with topotecan.

## 5. Conclusions

We studied the inhibition of DNA repair enzymes (PARP1, PARP2, and Tdp1) by lipophilic pyrimidine nucleosides. New effective Tdp1 inhibitors were found in a series of lipophilic pyrimidine nucleosides and have IC_50_ in the lower micromolar and submicromolar range. The 2,3,5-tri-*O*-Bz-β-d-Rib moiety appears to be the pharmacophore that is essential for Tdp1 inhibition by each synthesized compound. It was shown that the lipophilic pyrimidine nucleosides inhibiting Tdp1 in vitro have only low cytotoxicity. Some derivatives strengthen the action of topotecan on HeLa cells and thus may be considered promising for further optimization of the structure and the development of compounds sensitizing cancer cells to topotecan.

## 6. Patents

The application for RU patent №2019141448 was registered 13.12.2019.

## Figures and Tables

**Figure 1 molecules-25-03694-f001:**
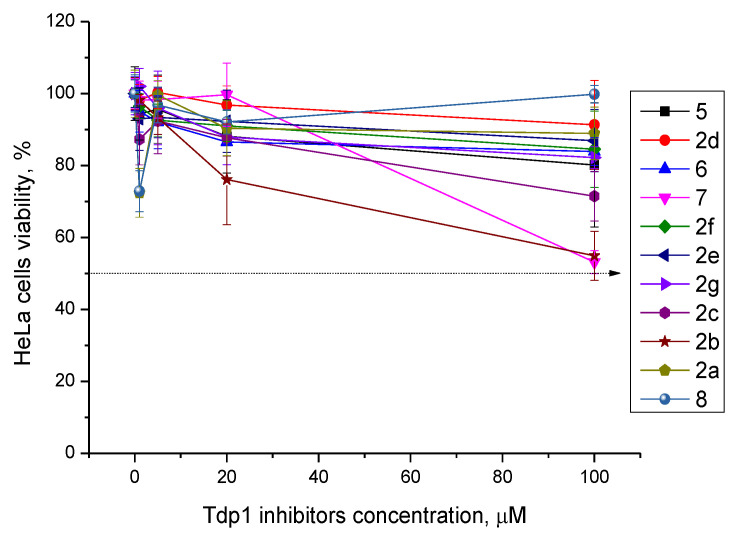
Dose-dependent action of the lipophilic pyrimidine nucleosides on HeLa cell viability measured in the EZ4U assay.

**Figure 2 molecules-25-03694-f002:**
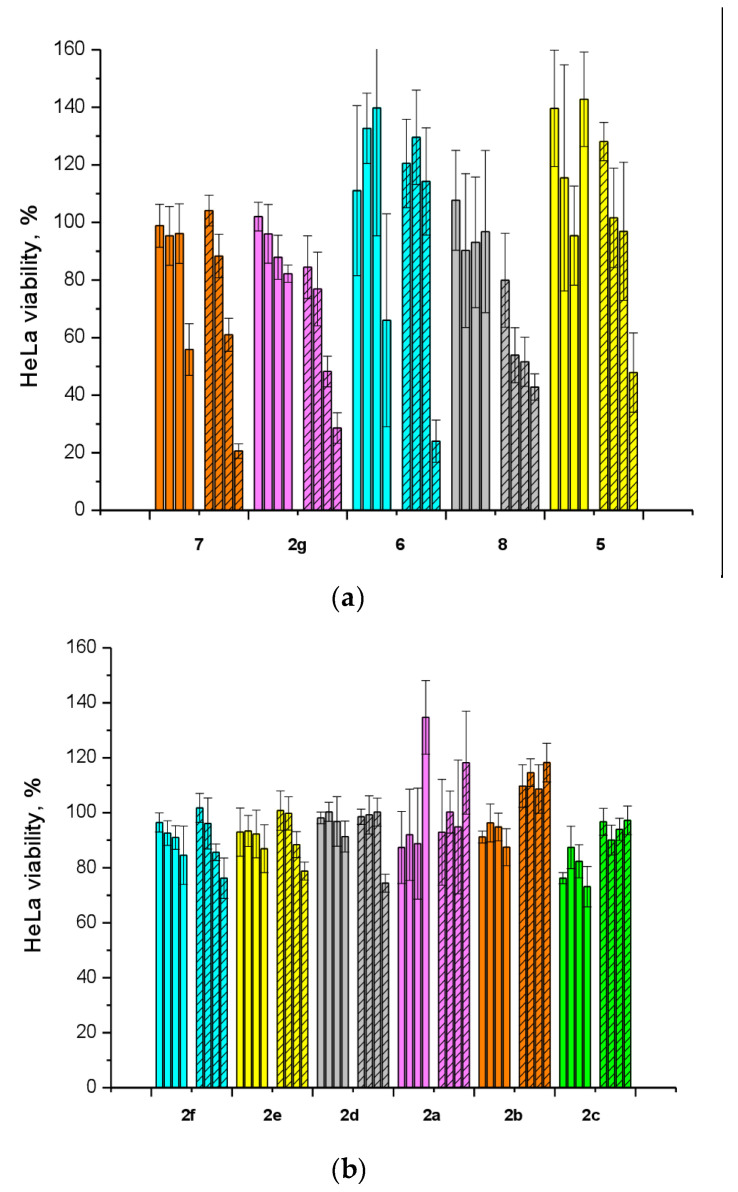
Dose-dependent action of the lipophilic nucleosides in combination with topotecan (Tpc) on HeLa cell viability in the EZ4U assay. The unshaded histogram bars denote cell viability in the presence of a single Tdp1 inhibitor. The hatched histogram bars indicate cell viability in the presence of a combination of a Tdp1 inhibitor with 2 μM topotecan. (**a**) Compounds showing synergy with topotecan. The colors indicate: orange–compound **7**, magenta—**2g**, cyan—**6**, gray—**8**, and yellow—**5**. (**b**) Inactive compounds. The colors indicate: cyan—compound **2f**, yellow—**2e**, gray—**2d**, magenta—**2a**, orange—**2b**, and green—**2c**. Each of the four bars corresponds to an inhibitor concentration (from left to right) of 1, 5, 20, and 100 μM.

**Figure 3 molecules-25-03694-f003:**
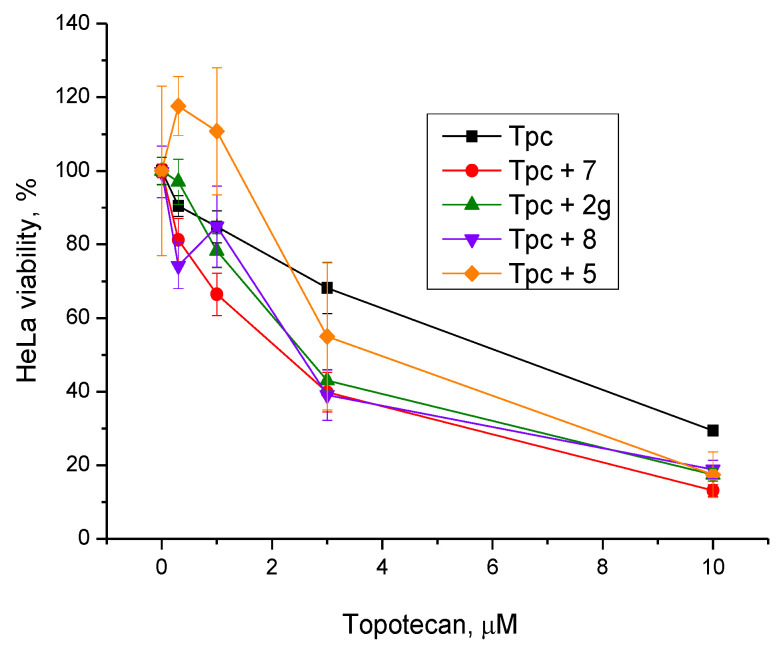
Dose-dependent action of topotecan (Tpc) in combination with one of the nucleoside derivatives on HeLa cell viability in the EZ4U assay.

**Figure 4 molecules-25-03694-f004:**
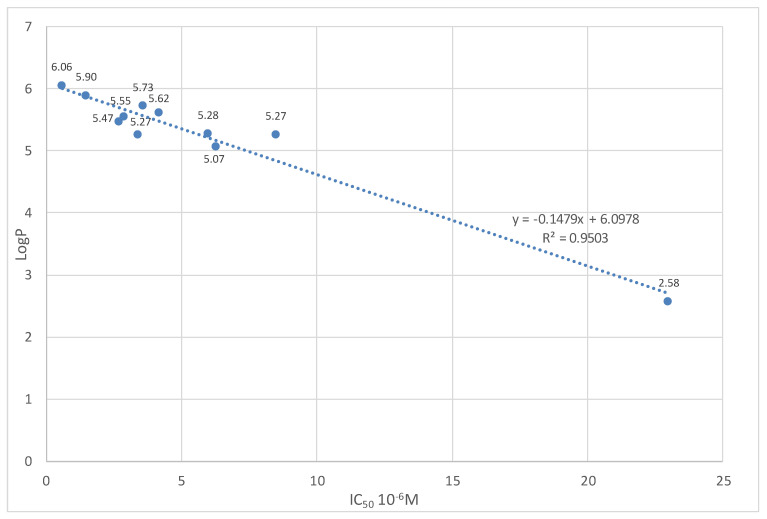
Correlation diagram between Tdp-1 inhibition IC_50_ and logP of nucleoside derivatives for compounds **2a**–**2g**, **4a**, **6**–**7** and **9** with exclusion of the data for compounds **5**, **8**.

**Figure 5 molecules-25-03694-f005:**
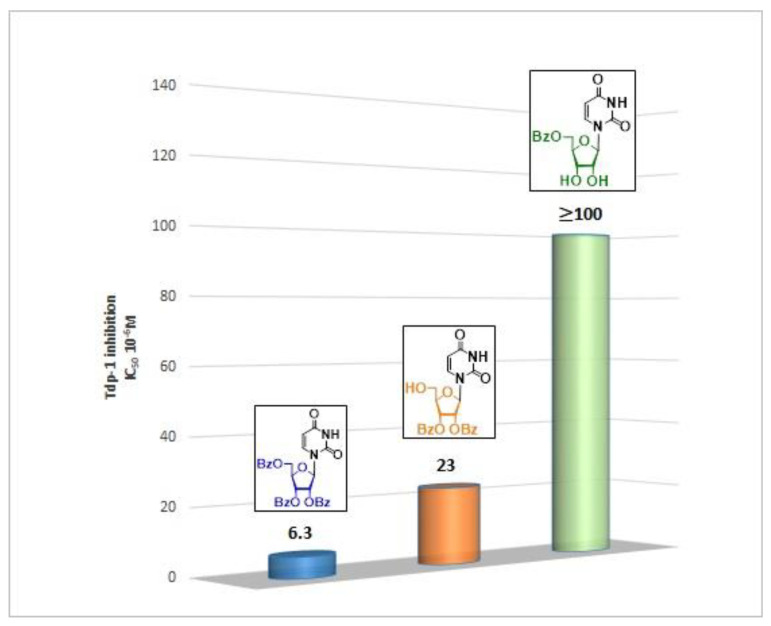
Structure-activity relationship between Tdp-1 inhibition and the quantity of benzoyl groups in nucleoside ribofuranose moiety.

**Table 1 molecules-25-03694-t001:** Inhibition of Tdp-1 by nucleoside derivatives.

Cmpd	Structure	LogP ^1^	IC_50_µM	HeLaCC_50_ µM
**1a**	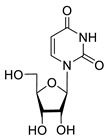	−2.28	>50	ND ^2^
**1b**	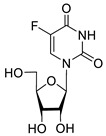	−2.64	>50	ND
**1f**	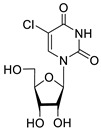	−2.24	>50	ND
**1c**	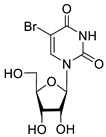	−1.97	>50	ND
**1d**	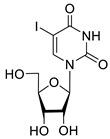	−1.44	>50	ND
**1e**	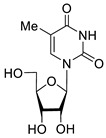	−1.39	>50	ND
**2a**	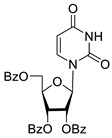	5.07	6.3 ± 0.4	ND
**2b**	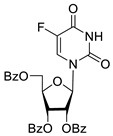	5.27	8.5 ± 1.4	>100
**2f**	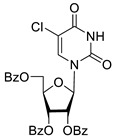	5.73	3.6 ± 1.1	>100
**2c**	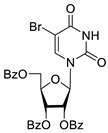	5.90	1.5 ± 0.9	>100
**2d**	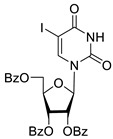	6.06	0.6 ± 0.9	>100
**2e**	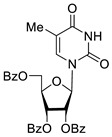	5.47	2.7 ± 0.6	>100
**2g**	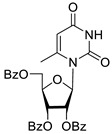	5.27	3.4 ± 0.2	>100
**3a**	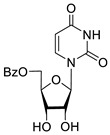	0.08	>100	ND
**4a**	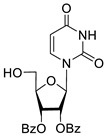	2.58	23 ± 6	>100
**5**	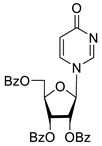	5.07	18 ± 1	ND
**6**	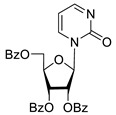	5.28	6.0 ± 0.7	>100
**7**	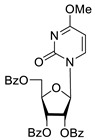	5.55	2.91 ± 0.01	>100
**8**	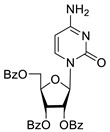	4.69	4.3 ± 0.7	>100
**9**	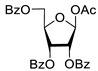	5.62	4.2 ± 0.3	ND

^1^ The values of the partition coefficient of the compounds between the 1-octanol-water phases (logP) were calculated using the Instant J. Chem. (ChemAxon^®^). ^2^ ND—not determined.

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
