# Peer review of "Inhibition of Tyrosyl-DNA Phosphodiesterase 1 by Lipophilic Pyrimidine Nucleosides"

_molecules, 2020, doi:10.3390/molecules25163694_

Round 1
Reviewer 1 Report
The work described herein reports the synthesis of a small set of perbenzoylated pyrimidine containing nucleoside analogues (mainly halogenated at the 5-position of uracil) and their biological evaluation as inhibitors of DNA repair enzymes (notably Tdp1).
The study (which is in line with ref 17) mainly refers to the biological evaluation of the compounds and should be submitted as a note or included in a wider project/manuscript. The chemistry part is lacking novelty and no new development concerning the biological assays are proposed in comparison to the previous research full paper by the same group (ref 17).
The chemistry section has to be removed and eventually include in the SI. Indeed, all compounds (2a-f) are already described in the literature, commercially available, and have been obtained by N-glycosylation. Therefore, extensive description of their synthesis and related NMR data in the result section is unnecessary.
In addition, the use of benzoyl cyanide for the perbenzoylation of nucleosides is known since 2005. The authors have omitted to cite the work of Prasad et al. in the result section, and ref 30 is only present in the exp. section, which is not fair.
The numbering in scheme 1, does not correspond to the compounds described in the exp section (2b is the fluoro derivative in the scheme, whereas 2b is the bromo derivative in the exp section, line 245). It is also misleading as in the text (lines 73-76) compounds 2b and 2e were obtained by glycosylation...
The discussion of the SAR is confusing and should be re-written considering as example the comparison of the same nucleoside with increasing number of Bz groups (2a, 3a, 4a) in relation with the logP variation, then the variation of the halogen in the 5-position.
Concerning the calculated logP values, in order to compare them it must be estimated with the same software as the values may differ from one calculator to another.
Finally, as benzoyl groups may be hydrolyzed in biological media (mediated by esterases) it would be very interesting to compare benzoylated and benzylated derivatives.
In the conclusion, the authors claimed that the 2,3,5-tri-benzoyl-b-D-ribofuranose moiety appeared as the pharmacophore group but did they tested the corresponding sugar moiety? such as the 2,3,5-tri-benzoyl-1-methoxy-ribofuranose or a-nucleoside analogues …
Author Response
Comments and Suggestions for Authors 1.
The work described herein reports the synthesis of a small set of perbenzoylated pyrimidine containing nucleoside analogues (mainly halogenated at the 5-position of uracil) and their biological evaluation as inhibitors of DNA repair enzymes (notably Tdp1).
The study (which is in line with ref 17) mainly refers to the biological evaluation of the compounds and should be submitted as a note or included in a wider project/manuscript. The chemistry part is lacking novelty and no new development concerning the biological assays are proposed in comparison to the previous research full paper by the same group (ref 17).
The chemistry section has to be removed and eventually include in the SI. Indeed, all compounds (2a-f) are already described in the literature, commercially available, and have been obtained by N-glycosylation. Therefore, extensive description of their synthesis and related NMR data in the result section is unnecessary.
Answer: We removed the chemistry section and related NMR spectra to SI.
In addition, the use of benzoyl cyanide for the perbenzoylation of nucleosides is known since 2005. The authors have omitted to cite the work of Prasad et al. in the result section, and ref 30 is only present in the exp. section, which is not fair.
Answer: We apologize, this is definitely not fair from our part. Therefore, we have eliminated this confusion. The paper published by Prasad and collegues was cited twice in the section 2.1 and in Experimental Part of the work (ref.26).
The numbering in scheme 1, does not correspond to the compounds described in the exp section (2b is the fluoro derivative in the scheme, whereas 2b is the bromo derivative in the exp section, line 245). It is also misleading as in the text (lines 73-76) compounds 2b and 2e were obtained by glycosylation...
Answer: Corrected.
The discussion of the SAR is confusing and should be re-written considering as example the comparison of the same nucleoside with increasing number of Bz groups (2a, 3a, 4a) in relation with the logP variation, then the variation of the halogen in the 5-position.
Answer: We corrected the discussion of SAR on the basis of logP variation and number of Bz groups.
Concerning the calculated logP values, in order to compare them it must be estimated with the same software as the values may differ from one calculator to another.
Answer: The logP values were calculated with Instant J Chem (ChemAxon software).
Finally, as benzoyl groups may be hydrolyzed in biological media (mediated by esterases) it would be very interesting to compare benzoylated and benzylated derivatives.
Answer: We did not considered benzylated nucleoside derivatives in this paper, because this is a theme of another paper, concerning with various synthetic compounds containing non-hydrolizable aliphatic and aromatic groups and their biological activity.
In the conclusion, the authors claimed that the 2,3,5-tri-benzoyl-b-D-ribofuranose moiety appeared as the pharmacophore group but did they tested the corresponding sugar moiety? such as the 2,3,5-tri-benzoyl-1-methoxy-ribofuranose or a-nucleoside analogues …
Answer: We did not tested a-nucleoside analogues, but the data for 2,3,5-tri-benzoyl-1-O-acetyl-ribofuranose was included into Table1. This compound manifested Tdp1 inhibition with IC50 = 4.2 micromoles/L, which is comparable with several nucleoside compounds described in table 1. Nucleoside derivatives with logP possessing higher lipophilicity manifested higher Tdp1-inhibition than 2,3,5-tri-benzoyl-1-O-acetyl-ribofuranose. The corresponding description was made in discussion of SAR (lines 147-154).
Reviewer 2 Report
In this study, the authors studied inhibition of DNA repair enzymes (PARP1, PARP2, Tdp1) by lipophilic pyrimidine nucleosides. New effective Tdp1 inhibitors were found in the series of lipophilic pyrimidine nucleosides with IC50 values in the lower micromolar and submicromolar range of concentrations. 2,3,5-Tri-О-benzoyl-β-D-ribofuranose moiety appeared to be the pharmacophore group, which was essential for inhibition of Tdp-1 by all the synthesized compounds. It was shown, that lipophilic pyrimidine nucleosides, inhibiting Tdp1 in vitro, manifested only low cytotoxicity and some derivatives strengthened the action of topotecan on HeLa cells. Overall, I think the work is interesting and well done. I have a minor comment about the discussion where the authors should include a section on the importance of understanding these repair enzyme in terms of human disease. Some useful reference for this include
Sonali Bhattacharjee and Saikat Nandi (2018) Rare Genetic Diseases with Defects in DNA Repair: Opportunities and Challenges in Orphan Drug Development for Targeted Cancer Therapy. Cancers. 10.3390/cancers10090298
Sonali Bhattacharjee and Saikat Nandi (2017) DNA damage response and cancer therapeutics through the lens of the Fanconi Anemia DNA repair pathway. Cell Communication and Signalling. 10.1186/s12964-017-0195-
Author Response
Comments and Suggestions for Authors 2
In this study, the authors studied inhibition of DNA repair enzymes (PARP1, PARP2, Tdp1) by lipophilic pyrimidine nucleosides. New effective Tdp1 inhibitors were found in the series of lipophilic pyrimidine nucleosides with IC50 values in the lower micromolar and submicromolar range of concentrations. 2,3,5-Tri-О-benzoyl-β-D-ribofuranose moiety appeared to be the pharmacophore group, which was essential for inhibition of Tdp-1 by all the synthesized compounds. It was shown, that lipophilic pyrimidine nucleosides, inhibiting Tdp1 in vitro, manifested only low cytotoxicity and some derivatives strengthened the action of topotecan on HeLa cells. Overall, I think the work is interesting and well done. I have a minor comment about the discussion where the authors should include a section on the importance of understanding these repair enzyme in terms of human disease. Some useful reference for this include
Sonali Bhattacharjee and Saikat Nandi (2018) Rare Genetic Diseases with Defects in DNA Repair: Opportunities and Challenges in Orphan Drug Development for Targeted Cancer Therapy. Cancers. 10.3390/cancers10090298
Sonali Bhattacharjee and Saikat Nandi (2017) DNA damage response and cancer therapeutics through the lens of the Fanconi Anemia DNA repair pathway. Cell Communication and Signalling. 10.1186/s12964-017-0195-
Answer: Much gratitudes for useful papers. We included these references into introduction section (ref.14-15), containing the discussion of importance of DNA repair enzymes and proteins.
Reviewer 3 Report
In the manuscript by Zakharenko et al., the authors screened lipophilic pyrimidine nucleosides (2’,3’,5’-tri-O-benzoyluridine derivatives and more) as potential enzymatic inhibitors of tyrosyl-DNA phosphodiesterase 1 (TDP1). The authors describe, that all benzoylated pyrimidine nucleosides had low micromolar (0.6-8.5 µM) IC50 values for TDP1 as judged by an in vitro assay while having low cytotoxicity towards HeLa cells in culture. Hence, the components were considered suitable as TDP1 candidate inhibitors for further investigations and optimizations. Moreover, three of the candidates enhanced the toxicity of Topotecan, a Topoisomerase inhibitor used for chemotherapy.
The results are interesting and of potential clinical relevance for cancer chemotherapy. I have, however, a number of issues that must be addressed in a revised version before I can recommend acceptance of the manuscript for publication.
Main issues:
- The title is misleading and should be changed since inhibitor screening is only presented against TDP1. A side-screen against PARP1 and -2 is mentioned but (i) is not presented and (ii) was not successful. No other repair enzymes were tested.
- The effect of component 6 to enhance the cytotoxicity of Topotecan displayed in Figure 2a was not reproducible (Figure 3). Why do the authors present and conclude on inconsistent data for component 6 instead of repeating the MTT assay to clarify its effect?
- Lines 133-134 “The cytotoxicity…did not significantly changed (Figure 2b).” Besides correcting the grammatical error, the authors should present calculated statistical significances in all three main figures before concluding about significant or non-significant changes.
- Components 2b, 7 and 2c are toxic to HeLa cells. The effect is neither “…absent or was negligible in concentrations up to 100 µM…” (lines 119-120).
- Component 1b is missing in Table 1 but mentioned in Scheme 1 and text (line 163).
- Supplementary Figures 15 (only left panel with all components) and 16 should be main figures. They nicely demonstrate a correlation between lipophilicity and efficacy of the components studied.
Minor issues:
- Grammatical errors and syntax throughout the text should be carefully corrected.
- The introduction (lines 33-48) is highly redundant and should be revised.
- The lines and symbols presented in Figures 1-3 are hard to distinguish and overlapping. Please consider a different graphical presentation of these results.
- Lines 165-179 should belong to the results section. It purely describes the results of the study. The aim of a discussion is not to repeat or substitute the results section.
- The authors should avoid using two different terms for the same assay: “EZ4U Cell Proliferation and Cytotoxicity Assay” (line 115) and “MTT assay” (line 123, and also Materials and Methods section).
Author Response
Comments and Suggestions for Authors 3
In the manuscript by Zakharenko et al., the authors screened lipophilic pyrimidine nucleosides (2’,3’,5’-tri-O-benzoyluridine derivatives and more) as potential enzymatic inhibitors of tyrosyl-DNA phosphodiesterase 1 (TDP1). The authors describe, that all benzoylated pyrimidine nucleosides had low micromolar (0.6-8.5 µM) IC50 values for TDP1 as judged by an in vitro assay while having low cytotoxicity towards HeLa cells in culture. Hence, the components were considered suitable as TDP1 candidate inhibitors for further investigations and optimizations. Moreover, three of the candidates enhanced the toxicity of Topotecan, a Topoisomerase inhibitor used for chemotherapy.
The results are interesting and of potential clinical relevance for cancer chemotherapy. I have, however, a number of issues that must be addressed in a revised version before I can recommend acceptance of the manuscript for publication.
Main issues:
The title is misleading and should be changed since inhibitor screening is only presented against TDP1. A side-screen against PARP1 and -2 is mentioned but (i) is not presented and (ii) was not successful. No other repair enzymes were tested.
Answer: Inhibition of tyrosyl-DNA phosphodiesterase 1 by lipophilic pyrimidine nucleosides?
The effect of component 6 to enhance the cytotoxicity of Topotecan displayed in Figure 2a was not reproducible (Figure 3). Why do the authors present and conclude on inconsistent data for component 6 instead of repeating the MTT assay to clarify its effect?
Answer: After the Mann-Whitney test, it was found that the effect of compound 6 on the topotecan effect was not significant. We noted this in the text of the article and removed the graph corresponding to compound 6 from Figure 3.
Lines 133-134 “The cytotoxicity…did not significantly changed (Figure 2b).” Besides correcting the grammatical error, the authors should present calculated statistical significances in all three main figures before concluding about significant or non-significant changes.
Answer: Thank you for the advice. We applied the Mann-Whitney test to assess the significance of differences between the cytotoxicity values at each concentration point in pairs in the presence of one substance and a combination of substances.
Components 2b, 7 and 2c are toxic to HeLa cells. The effect is neither “…absent or was negligible in concentrations up to 100 µM…” (lines 119-120).
Answer: Indeed, these three compounds are moderately toxic at a concentration of 100 μM. We have added the phrase (lines 96-97): “with the exception of compounds 2b, 2c, and 7, in the presence of which at a concentration of 100 μM, 30-45% of the cells died”.
Component 1b is missing in Table 1 but mentioned in Scheme 1 and text (line 163).
Answer: We added compound 1b to the Table 1
Supplementary Figures 15 (only left panel with all components) and 16 should be main figures. They nicely demonstrate a correlation between lipophilicity and efficacy of the components studied.
Answer: Figures 15 and 16 was moved to the main text
Minor issues:
Grammatical errors and syntax throughout the text should be carefully corrected.
Answer: We have tried to thoroughly correct the grammar and syntax.
The introduction (lines 33-48) is highly redundant and should be revised.
Answer: The text of introduction (lines 33-48) was revised and shortened (the corrections are marked with green).
Answer: We tried to correct grammatical errors
The lines and symbols presented in Figures 1-3 are hard to distinguish and overlapping. Please consider a different graphical presentation of these results.
Answer: The lines and symbols in Figures 1 and 3 have been made brighter. In Figure 2, the graphs were replaced by histograms. We hope that in this form, the drawings are easier to read.
Lines 165-179 should belong to the results section. It purely describes the results of the study. The aim of a discussion is not to repeat or substitute the results section.
Answer: We have re-written the text according to the recommendations of reviewer 1, so it has become more compatible with the discussion section.
The authors should avoid using two different terms for the same assay: “EZ4U Cell Proliferation and Cytotoxicity Assay” (line 115) and “MTT assay” (line 123, and also Materials and Methods section).
Answer: We used EZ4U test, so it was indicated everywhere in the text.
Round 2
Reviewer 3 Report
Thank you for changing the manuscript according to my comments.
1. The title is fitting better now.
2. I would recommend to shrink Figure 4 down to one panel: Panel b as main figure and panel a (with all components and the lower Rsquare) to the supplement.
Author Response
Thank you very much for the recommendation, we made the corresponding changes for Figure 4 in the main text and supplementary.